# Coherent manipulation of a solid-state artificial atom with few photons

V. Giesz[1,*], N. Somaschi[1,*], G. Hornecker[2,3,*], T. Grange[2,3], B. Reznychenko[2,3], L. De Santis[1,4], J. Demory[1], C. Gomez[1], I. Sagnes[1], A. Lemaître[1], O. Krebs[1], N.D. Lanzillotti-Kimura[1], L. Lanco[1,5], A. Auffeves[2,3] & P. Senellart[1,6]

In a quantum network based on atoms and photons, a single atom should control the photon state and, reciprocally, a single photon should allow the coherent manipulation of the atom. Both operations require controlling the atom environment and developing efficient atom–photon interfaces, for instance by coupling the natural or artificial atom to cavities. So far, much attention has been drown on manipulating the light field with atomic transitions, recently at the few-photon limit. Here we report on the reciprocal operation and demonstrate the coherent manipulation of an artificial atom by few photons. We study a quantum dot-cavity system with a record cooperativity of 13. Incident photons interact with the atom with probability 0.95, which radiates back in the cavity mode with probability 0.96. Inversion of the atomic transition is achieved for 3.8 photons on average, showing that our artificial atom performs as if fully isolated from the solid-state environment.

[1] CNRS-LPN Laboratoire de Photonique et de Nanostructures, Université Paris-Saclay, Route de Nozay, 91460 Marcoussis, France. [2] Université Grenoble Alpes, F-38000 Grenoble, France. [3] CNRS, Institut Néel, Nanophysique et Semiconducteurs Group, F-38000 Grenoble, France. [4] Université Paris-Sud, Université Paris-Saclay, F-91405 Orsay, France. [5] Département de Physique, Université Paris Diderot, 4 rue Elsa Morante, 75013 Paris, France. [6] Département de Physique, Ecole Polytechnique, Université Paris-Saclay, F-91128 Palaiseau, France. * These authors contributed equally to this work. Correspondence and requests for materials should be addressed to A.A. (email: Alexia.Auffeves@neel.cnrs.fr) or to P.S. (email: Pascale.Senellart@lpn.cnrs.fr).

The light-matter interaction provides a natural framework to interface photonic channels and quantum nodes[1] based on atoms[2,3], ions[4] or artificial atoms, such as semiconductor quantum dots (QDs)[5–8] or defects in diamonds[9]. However, mapping a photonic channel onto a single (artificial) atom in free space is naturally inefficient[10]. An ideal atom–photon interface requires that a single atom interacts with only a single and well-defined mode of the optical field. Such ideal one-dimensional (1D) atom systems are developed by engineering the electromagnetic field around the natural or artificial atom[11,12] using photonic structures like single-sided leaky cavities[13–17], single-mode photonic waveguides[18,19] or fibres evanescently coupled to directional cavity modes[20]: the spontaneous emission is accelerated into a well-defined mode of the electromagnetic field[13–17,20] or inhibited in all other modes[18,19]. Moreover, an efficient interface requires a coherent response from the atom that should be well isolated from any source of decoherence, a major challenge for artificial atoms.

Various quantum operations can be envisioned with an atom–photon interface: the atomic transition can be used to manipulate incoming photons or, reciprocally, an incoming photon can coherently manipulate the atomic state. So far most efforts have focused on the photon control by the atom, making use of the atom ability to scatter only one photon at a time[5–8,14,15,20–22], thus controlling the phase and photon statistics of the reflected or transmitted optical field. The relevant figure of merit of such optical gate is the absolute number of photons sent on the device. Operation of the gates with few incident photons has very recently been demonstrated in atomic systems[14,20,22], whereas, for QD-based systems, hundred or thousands of photons need to be sent onto the device because of a poor in-coupling efficiency[5–7,19,21] or a strong dephasing of the atomic transition[8].

The reciprocal operation, where a single incident photon manipulates the atom, is also very attractive as it holds the promise to efficiently and coherently transfer the information from the photon to the atomic quantum node[1]. This requires not only an efficient interface, but also no source of decoherence and a way to monitor the atom state during the interaction. So far, apart from pioneer results obtained with natural atoms in macroscopic cavities[23], the demonstration of coherent manipulation of an atom by few photons has remained elusive. In the present work, we demonstrate the coherent manipulation of a QD-based artificial atom with a $\pi$-pulse obtained for 3.8 incident photons on the device. These results are obtained on an electrically controlled QD pillar cavity device[24] providing a close to ideal 1D atom situation and a complete suppression of pure dephasing phenomena.

## Results

**An efficient photon–atom interface**. We study the excitonic transition of a semiconductor QD coupled to a microcavity. Light is confined vertically in a $\lambda$-cavity surrounded by two GaAs/Al$_{0.9}$Ga$_{0.1}$As Bragg-reflectors and laterally through the etching of a connected pillar structure as introduced in ref. 25. Figure 1a presents a sketch of the device: a single QD is inserted in a circular micropillar cavity with a diameter of 2.9 μm, connected through 1D 1.4 μm wide ridges to a large frame where an electrical contact is defined. Combined with a p-i-n doping of the layers, this geometry allows applying a bias to the structure, a critical tool to finely tune the QD resonance to the cavity mode energy as well as to stabilize the charge environment of the QD[24]. The cavity is fabricated using a cryogenic *in situ* lithography to centre each cavity device on a single QD with 50 nm spatial accuracy[26]. Evidence for this deterministic positioning is provided by the emission map of the device shown in Fig. 1b: a

very bright signal originating from the QD appears at the centre of the cavity, signature of an increased collection efficiency.

The sample is placed in a closed-cycle cryostat at 4 K and a confocal geometry setup is used to excite the QD and collect the signal through a microscope objective (NA 0.75). Figure 1c presents emission spectra of the device at zero bias under non-resonant excitation at 850 nm, collected in two linear polarizations labelled $H$ and $V$. Each spectrum presents two emission lines, one corresponding to the cavity mode and the other to the QD exciton line. Both line energies depend on the detected polarization. The fundamental mode of the pillar around 1.3399 eV decomposes into two linearly polarized modes ($H$ and $V$) split by 70 μeV with a linewidth of $\kappa = 120$ μeV (quality factors around 11,000) ensuring a good spectral overlap between both modes whereas the exciton fine structure splitting (FSS) amounts to $\Delta_{\mathrm{FSS}} = 15$ μeV. As shown by the polar plot in Fig. 1d, the QD axes labelled $X$ and $Y$ are at 40° from the cavity ones, a configuration naturally resulting from the cavity design where the connecting wires were intentionally defined at ~45° from the crystal axes. This configuration allows us to monitor the exciton population using resonant fluorescence as explained below.

To characterize the QD–cavity interface, reflectivity measurements are performed scanning a $V$-polarized continuous wave laser across the cavity mode resonance. Figure 1e, obtained for an applied bias corresponding to the cavity-exciton resonance, evidences a strong coherent response of the QD at the centre of the cavity dip[8]. Using a $V$-polarized excitation and collection, only the $V$ cavity mode is visible while both exciton dipoles polarized along $X$ and $Y$ are excited, giving rise to a slightly asymmetric QD-induced reflectivity peak. A theoretical adjustment shown in red, which considers the two exciton states and the two cavity modes (Supplementary Note 1), evidences a cavity-QD coupling strength $g = 21$ μeV. From this adjustment, we also deduce that the exciton line is actually radiatively limited, with a dephasing rate of $\gamma = 0.3$ μeV. Such linewidth deduced from a time average reflectivity measurement demonstrates a strong suppression of charge noise in the gated structures over a hundred of milliseconds time scale, an observation consistent with the emission of long streams of highly indistinguishable photons with similar devices[24,27]. Such low dephasing results in a cooperativity as large as $C = \frac{g^2}{\kappa\gamma} = 13$, similar or greater than the one typically observed in natural atom systems[14,15,20,22].

At high excitation power, the QD response saturates and only the reflectivity dip of the bare cavity is visible (Supplementary Fig. 1). The minimum reflectivity then provides the fraction of photons escaping the cavity mode through the top mirror, that is, the out-coupling efficiency, $\eta_{\mathrm{out}} = 0.7 \pm 0.05$. Finally, we measure the mode profile of the incoming beam and that of the pillar fundamental mode and deduce an excellent mode matching corresponding to an input coupling efficiency exceeding $\eta_{\mathrm{in}} > 0.95$. These measurements show that the present device is close to the textbook situation of the 1D atom obtained here for a solid-state artificial atom. Every photon sent on the device enters the cavity with probability $\eta_{\mathrm{in}} > 0.95$ and interacts with the exciton. The exciton then radiates back into the cavity mode with a probability $\frac{2C}{2C+1} = 0.96$ and the emitted photon escapes through the top mirror with a probability $\eta_{\mathrm{out}} = 0.7 \pm 0.05$. Such near ideal 1D atom configuration and the negligible pure dephasing of the transition put our system in a unique position to coherently transfer the quantum state from the optical field to the atom at the very few photon level as discussed now.

**Probing the atom state**. The artificial atom state is probed by resonant fluorescence measurements in a crossed polarization

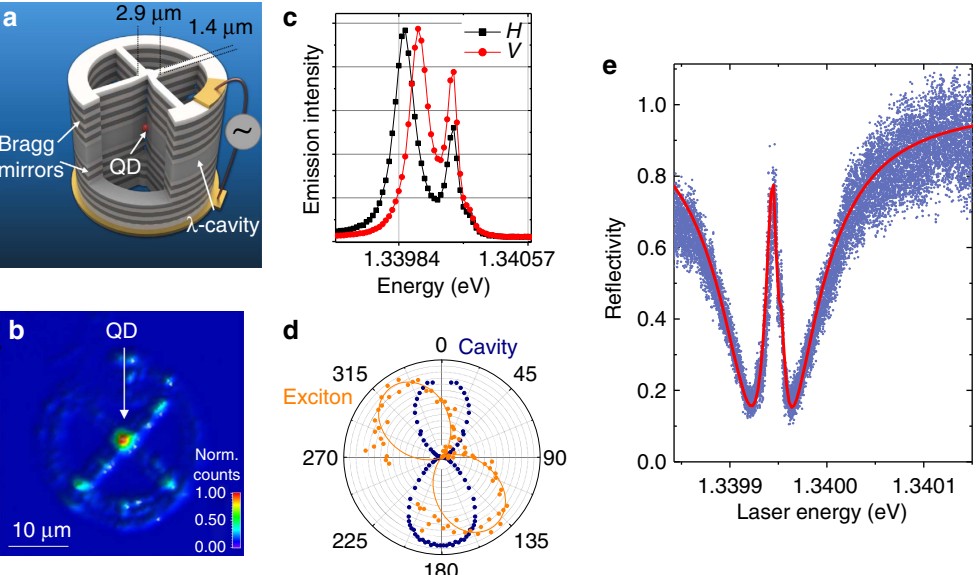

**Figure 1 | A quantum dot in a cavity as a one-dimensional atom system.** (**a**) Sketch of the device: a single QD is located at the centre of a pillar cavity connected through 1.4 μm wide ridges to a circular frame where the electrical contact is defined. (**b**) Emission map of the device obtained by scanning the sample below the excitation laser spot. (**c**) Emission spectra measured at 0 V bias under non-resonant excitation at 850 nm for two linear polarizations for the detection, labelled H and V. (**d**) Polar plot of the QD exciton energy (orange) and of the cavity mode energy (blue) deduced from polarization resolved emission measurements (see **c**). (**e**) Reflectivity measured for an excitation power around 50 pW by scanning a V-polarized monomode continuous wave laser across the cavity resonance at the bias of resonance with the QD. Blue: measured reflectivity; Red: theoretical fit (Supplementary Fig. 1 and Supplementary Note 2).

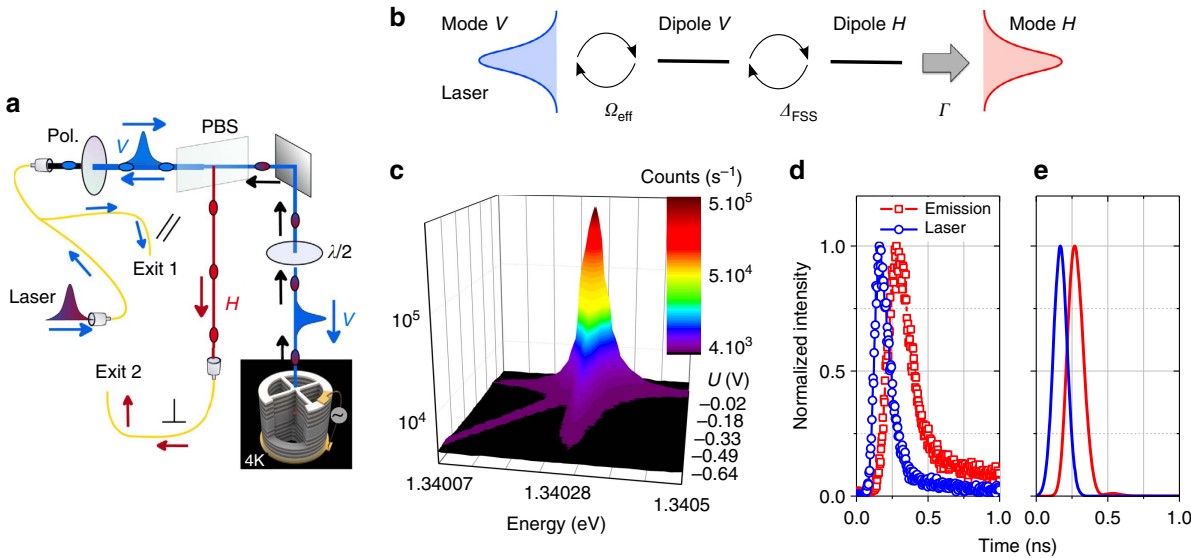

**Figure 2 | Resonant fluorescence measurement used to monitor the exciton population.** (**a**) Sketch of the experimental setup: the laser polarization is controlled by a polarizing beam splitter and a half waveplate. The reflectivity is measured by collecting the signal at exit 1 while the resonant fluorescence is measured at exit 2 in crossed polarization. (**b**) Schematic representation of the theoretical model used to describe the experiment. See text for details. (**c**) Resonant fluorescence intensity (log. scale) measured in crossed polarization as a function of energy and applied bias. The laser is resonant to the V mode energy. (**d**,**e**) Time dependence of the 56 ps excitation pulse (blue) and the collected emission in H polarization (red). **d**, experiment; **e**, calculations.

detection scheme. The device is excited with optical pulses arising from a Titanium-Sapphire laser delivering 3 ps pulses at 82 MHz repetition rate. These pulses are shaped with a spectrometer and an etalon to obtain quasi-gaussian pulses with durations between 10 and 90 ps. We excite the device with linearly polarized light along V and separate the H and V-polarized optical responses

(Fig. 2a). The exciton population is monitored by measuring the exciton fluorescence in the orthogonal H polarization.

Figure 2c presents the resonant fluorescence emission collected in H polarization when continuously varying the bias applied on the device. The laser energy is set to the energy of the V-polarized cavity mode. By increasing the applied voltage, the QD excitonic

transition is tuned in resonance with the laser energy where a strong fluorescence signal is observed. Away from resonance, a faint emission at the exciton energy is still observed due to phonon-assisted processes. The mechanism leading to an exciton emission in crossed polarization can be seen as follows: the laser drives the $V$-polarized exciton state $|\psi(t=0)\rangle=|V\rangle$, which corresponds to a linear superposition of $X$ and $Y$ exciton states: $|V\rangle=\frac{|X\rangle+|Y\rangle}{\sqrt{2}}$. This superposition temporally evolves as $|\psi(t)\rangle=\frac{|X\rangle+e^{-i\Delta_{FSS}t/\hbar}|Y\rangle}{\sqrt{2}}$, eventually leading to an emission along $H$ polarization. To theoretically account for the experiment, a model schematically described in Fig. 2b and presented in details in the Supplementary Fig. 2 and Supplementary Note 2 is developed. The QD is modelled as a $V$-type 3-level system (that is, the ground state $|g\rangle$ and the two polarized excitons $|X\rangle$ and $|Y\rangle$), neglecting multi-excitonic states). Its interaction with the $V$ and $H$-polarized cavity modes is treated within the rotating wave approximation. When the $V$-polarized cavity mode is driven by a coherent field, the density matrix of the system consisting of the $V$-polarized mode coupled to the QD is calculated using a Lindblad master equation (see Supplemental Materials for details). Figure 2d,e present the measured and calculated time dependences of the 56 ps excitation pulse (blue) and the $H$-polarized exciton emission (red). The emission is temporally delayed from the excitation pulse in agreement with theoretical predictions: the smaller the fine structure splitting, the longer the temporal evolution toward the orthogonal exciton state.

**Coherent control measurements**. We now study the coherent control of the exciton by monitoring the time integrated $H$-polarized emission intensity. The exciton population oscillates during the pulse, driven by the $V$-polarized excitation. The

$H$-polarized emission mostly takes place after the pulse as shown in Fig. 2d. When increasing the average power for a given pulse duration, the probability for the QD to be in the exciton state at the end of the pulse oscillates[28,29], resulting in oscillations in the $H$-polarized emission with the excitation power. Figure 3a presents the time integrated $H$-polarized emission intensity as a function of the average power sent on the device (top scale). Clear Rabi oscillations are evidenced for both 12 and 56 ps pulses. The bottom scale of Fig. 3a shows the mean photon number $\langle n\rangle$ per pulse sent on the device, derived from the average excitation power $P$, the laser repetition rate $1/\Gamma_{rep}=12$ ns and the photon energy $E=1.34$ eV: $\langle n\rangle=\frac{P}{\Gamma_{rep}E}$. Only 3.8 (8.6) photons are sent on average at $\pi$-pulses for a pulse duration of 56 (12) ps. This extremely low photon number is obtained owing to the excellent mode matching between the incident gaussian beam and the fundamental mode of the micropillar. Experimental observations are theoretically reproduced in Fig. 3b presenting the calculated $H$-polarized exciton emission as a function of $\langle n\rangle$. The model includes only the parameters extracted from Fig. 1e. It assumes a perfect input coupling efficiency ($\eta_{in}=1$) and no pure dephasing of the exciton transition. An excellent agreement with the experimental observations is obtained. Although these Rabi oscillations demonstrate the few-photon coherent control of the exciton, they do not provide a direct measurement of the probability to invert the exciton state. This flip probability can be derived from the theoretical model by calculating the probability of finding the QD in its excited state (both $H$ or $V$ dipole) as shown in Fig. 3c for the two pulse durations. At $\pi$-pulse, sending 8.6 (3.8) photons is sufficient to flip the exciton state with a probability of 81% (62%) depending on the pulse length. The flip probability is slightly reduced for the longer pulse: Rabi oscillations driven by the $V$ excitation are damped by the spontaneous emission of the exciton in the $V$ mode and by the

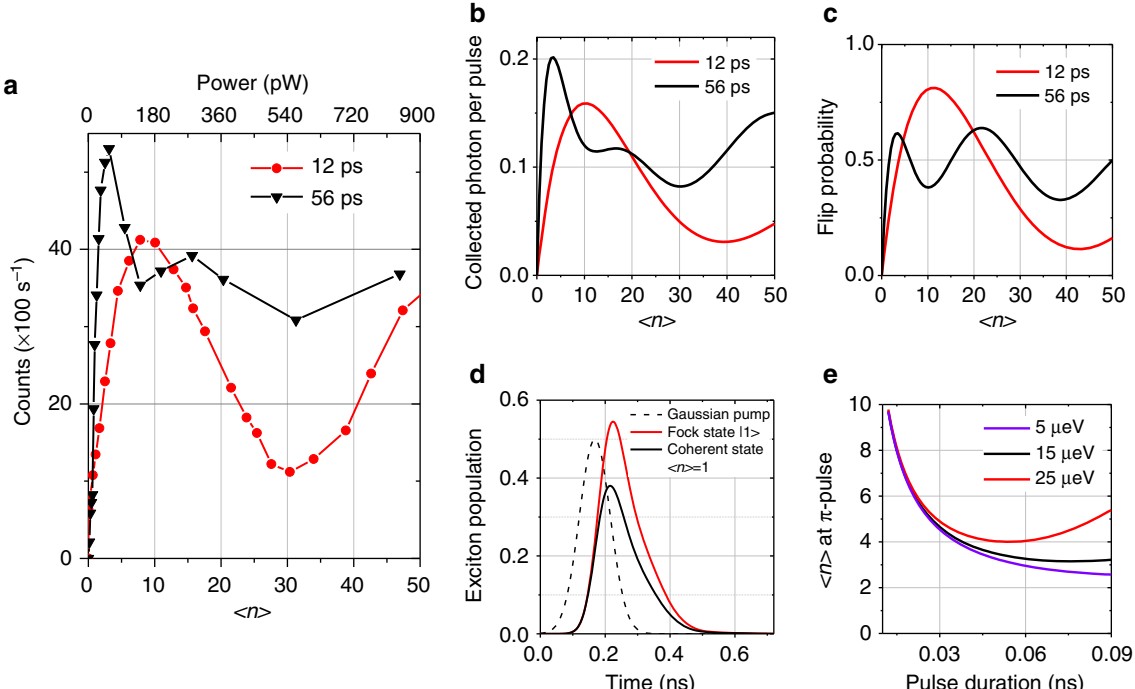

**Figure 3 | Coherent control with few-photon pulses.** (**a**) $H$-polarized emission intensity as a function of the excitation power (top axis) and the average photon number $\langle n\rangle$ (bottom axis) for two pulse durations (12 and 56 ps). (**b**) Calculated $H$-polarized emission as a function of $\langle n\rangle$ for two pulse durations. (**c**) Calculated probability to find the QD in its excited state after the pulse as a function of $\langle n\rangle$ for 12 and 56 ps pulses. (**d**) Calculated probability to find the QD in the exciton state as a function of time for a 56 ps excitation pulse (dashed line). Black: case of a coherent pulse with $\langle n\rangle=1$. Red: case of a single photon Fock state. (**e**) Mean photon number $\langle n\rangle$ needed to induce a $\pi$-pulse as a function of the pulse duration for various values of $\Delta_{FSS}$.

   

time evolution toward the $|H\rangle$ exciton state. With a Purcell factor of 13, corresponding to an emission decay time around 100 ps, spontaneous emission damping hardly plays a role for the 12 ps pulse, while oscillations are damped for a 56 ps pulse. On the other hand, more photons are needed with the 12 ps pulse, because of the lower spectral overlap between the excitation pulse and the exciton state.

## Discussion

To the best of our knowledge, the coherent control of an artificial atom has never been demonstrated at the very few-photon level. Few works reported on the coherent control of a QD transition in a cavity[21,30], but with a poor efficiency because of the poor overlap between the incident field and the photonic crystal cavity mode. A non-linearity at the few incident photon level was measured for a QD in a pillar cavity, with an onset of the non-linearity at the level of eight photons per pulse[8]. However, population inversion was out of reach and the saturation of the exciton transition was obtained for around a thousand photons. This was due to a limited output coupling efficiency and to a significant QD dephasing which prevented the coherent manipulation of the QD transition. An efficient atom manipulation with few single photons has only been demonstrated with a single natural atom trapped in a cavity[23]. It is demonstrated here for a micron-size semiconductor device where the artificial atom acts as fully isolated from solid-state environment.

To flip the exciton state with even lower photon numbers, two parameters should be considered. First, the current experiment uses attenuated coherent pulses: for a coherent pulse with $\langle n \rangle = 1$, the probability that the pulse actually contains no photons is $1/e = 36\%$. Figure 3d presents the population of the QD excited state as a function of time comparing the case of an incident coherent state with $\langle n \rangle = 1$ and the case of a single photon Fock state. To account for an incoming single photon wavepacket, the generalized master equation of (ref. 31) is used (Supplementary Note 2). A 44% increase of the probability to flip the exciton is expected: sending a single photon on the same device would ensure a probability to flip the atomic state with 55% probability. The other parameter that increases the number of photons at $\pi$-pulse is specific to the present experiment. Since the exciton state presents a sizeable FSS, around $\Delta_{FSS} = 15\,\mu eV$, the $V$ exciton dipole rotates towards the $H$ exciton dipole where efficient spontaneous emission into the $H$ cavity mode takes place. This mechanism contributes to damp the exciton population during the excitation, leading to a higher $\langle n \rangle$ at $\pi$-pulse. Figure 3e presents the average number of photon at $\pi$-pulse for three values of $\Delta_{FSS}$, as a function of the pulse duration. For each value of the $\Delta_{FSS}$, the mean photon number $\langle n \rangle$ at $\pi$-pulse presents a minimum value. Smaller the $\Delta_{FSS}$, lower the minimal photon number $\langle n \rangle$ is and longer the optimal pulse. Note that all curves merge for short pulses because of the spectral filtering by the cavity mode.

We have reported on a highly efficient solid-state interface between a photon and an artificial atom. This has been obtained by deterministically inserting a QD in an electrically controlled pillar cavity that ensures both an excellent mode matching with incident Gaussian beams and a strong suppression of pure dephasing phenomena. The device presented here has been designed to operate with a neutral exciton. Two types of quantum bits are explored with single QDs: excitons[32] or spin quantum bits[33], the latter offering the possibility to store the quantum information on microsecond time scales. The reported results can be extended to manipulate a charged exciton resonance allowing an efficient transfer of the quantum information between flying quantum bits (polarized photons) and a quantum memory (spin)[34,35]. Such efficient interface would open the way to deterministic quantum gates between delayed photons, and to the generation of photonic cluster states[36].

**Data availability**. All relevant data are available on request.

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

## Acknowledgements

This work was partially supported by the ERC Starting Grant No. 277885 QD-CQED, the French Agence Nationale pour la Recherche (grant ANR QDOM and SPIQE) the French RENATECH network, the Labex NanoSaclay ANR-10-LABX-0035, and the EU FP7 Grant No. 618072 (WASPS), N.D.L.-K. was supported by the FP7 Marie Curie Fellowship OMSiQuD.

## Author contributions

All authors substantially contributed to this work. The measurements were conducted primarily by V.G. and N.S. The sample was grown by C.G. and A.L., and the etching performed by I.S. The deterministic fabrication of the devices was done by N.S. Theory was developed by G.H., T.G., B. R. and A.A. The project was conducted by P.S. on the experimental side and by A.A. on the theory side.

## Additional information

**Competing financial interests:** The authors declare no competing financial interests.

