## [Peer review file · Nature Communications]

Transferred manuscripts: Editorial Note: this manuscript has been previously reviewed at another journal that is not operating a transparent peer review scheme. This document only contains reviewer comments and rebuttal letters for versions considered at Nature Communications.

Reviewers' Comments:

Reviewer #1 (Remarks to the Author)

The paper discussed the coherent control of a QD exciton inserted with deterministic location in a micro-pillar cavity. By electric bias of a p-i-n junction the exciton transition is tuned at resonance to the cavity mode energy, leading to a large cooperativity $C=13$ and to impressive out-and inputcoupling efficiencies of 0.7 and 0.95, respectively. Exploiting these relevant achievements, the authors report the observation of exciton Rabi flipping by means of 3.8 photons. The experimental data are convincing and the discussion is sound and clear. Even if the results could be expected by the technological expertise the French group has developed in pillar cavities in the last few year, the paper discuss new achievements. Indeed this is the first experimental demonstration of coherent control of a quantum transitions by few photons pulse. The paper deserves publication in Nature Communications after discussing two minor points:

1) The cavity has a polarization splitting of 0.07 meV. The exciton has a FSS of 0.025 meV. The two axis are rotated by 40 degrees. It is not stated, but it looks like these features are unintentional. I wonder if they are a problem and if the problem can be fixed in some way. This point needs to be discussed.

2) Multiexcitons usually play a role in the QD recombination kinetics. Are biexciton and charged exciton involved in the experiment? The point is not discussed.

Reviewer #3 (Remarks to the Author)

The manuscript by Giesz, Somaschi, Hornecker et al. reports on the coherent control of a quantum dot exciton transition coupled to a specifically designed micropillar cavity. In particular, by sending laser pulses with very low average photon numbers (down to 3.8) on the light-matter interface, the authors demonstrate a full π -phase shift of the QD exciton. This is a very impressive achievement and demonstrates strong progress towards the realization of elementary quantum networks in the solid state. I most certainly recommend publication in Nature Communications.

I should also stress that as opposed to the initial version under review for Nature Materials, the revised version of the paper is very clear about the impact and guides the reader very well to the key result of the paper by putting it in context with the existing literature on the topic. The authors have also adequately addressed the minor comments I previously made.

Response to reviewers

REVIEWERS' REQUESTS: Response to Reviewer #1 (Remarks to the Author): The paper discussed the coherent control of a QD exciton inserted with deterministic location in a micro-pillar cavity. By electric bias of a p-i-n junction the exciton transition is tuned at resonance to the cavity mode energy, leading to a large cooperativity $C=13$ and to impressive out-and input- coupling efficiencies of 0.7 and 0.95, respectively. Exploiting these relevant achievements, the authors report the observation of exciton Rabi flipping by means of 3.8 photons. The experimental data are convincing and the discussion is sound and clear. Even if the results could be expected by the technological expertise the French group has developed in pillar cavities in the last few year, the paper discuss new achievements. Indeed this is the first experimental demonstration of coherent control of a quantum transitions by few photons pulse. The paper deserves publication in Nature Communications after discussing two minor points: 1) The cavity has a polarization splitting of 0.07 meV. The exciton has a FSS of 0.025 meV. The two axis are rotated by 40 degrees. It is not stated, but it looks like these features are unintentional. I wonder if they are a problem and if the problem can be fixed in some way. This point needs to be discussed.

The cavity modes result from some a small anisotropy coming from the 1D wires connected to the pillars. The cavity mode splitting would be a problem if it were large as compared to the mode linewidth, which is not the case here where a good spectral overlap of both linearly polarized modes is maintained. The QD axes are usually roughly aligned to the crystal axes. As a result, by defining the cavity wires 45° from the crystal axes, we made sure that the QD and cavity axes are roughly 45° from each other's (close to the 40° measured value here). We have added a sentence to explain this. The influence of the FSS on our measurement is studied in details in figure 3

2) Multiexcitons usually play a role in the QD recombination kinetics. Are biexciton and charged exciton involved in the experiment? The point is not discussed.

Multiexcitons only play a role in non-resonant excitation schemes. Here, using strictly resonant excitation, one can neglect the high energy multi-exciton states. We have added a note to precise this assumption. This approximation is totally justified as attested by the excellent agreement between measurement and theory.